# Association of Renin-Angiotensin System Blockers with Survival in Patients on Maintenance Hemodialysis

**DOI:** 10.3390/jcm12093301

**Published:** 2023-05-05

**Authors:** Seok Hui Kang, Bo Yeon Kim, Eun Jung Son, Gui Ok Kim, Jun Young Do

**Affiliations:** 1Division of Nephrology, Department of Internal Medicine, College of Medicine, Yeungnam University, Daegu 42415, Republic of Korea; kangkang@ynu.ac.kr; 2Healthcare Review and Assessment Committee, Health Insurance Review and Assessment Service, Wonju 26465, Republic of Korea; 3Quality Assessment Department, Health Insurance Review and Assessment Service, Wonju 26465, Republic of Korea

**Keywords:** hemodialysis, renin-angiotensin system, survival, angiotensin-converting enzyme inhibitor, angiotensin II receptor blocker

## Abstract

Additional studies are needed to confirm whether the use of renin-angiotensin system blockers (RASBs) induces survival benefits in patients on hemodialysis (HD). This study aimed to evaluate patient survival with the use of RASBs in a large sample of maintenance HD patients. This study used data from the national HD quality assessment program and claim data from South Korea (*n* = 54,903). A patient using RASBs was defined as someone who had received more than one prescription during the 6 months of each HD quality assessment period. The patients were divided into three groups as follows: Group 1, no prescription for anti-hypertensive drugs; Group 2, prescription for anti-hypertensive drugs other than RASBs; and Group 3, prescription for RASBs. The five-year survival rates in Groups 1, 2, and 3 were 72.1%, 64.5%, and 66.6%, respectively (*p* < 0.001 for Group 1 vs. Group 2 or 3; *p* = 0.001 for Group 2 vs. Group 3). Group 1 had the highest patient survival rates among the three groups, and Group 3 had higher patient survival rates compared to Group 2. Group 3 had higher patient survival rates than Group 2; however, the difference in patient survival rates between Group 2 and Group 3 was relatively small. Multivariate Cox regression analyses showed similar trends as those of univariate analyses. The highest survival rates from our study were those of patients who had not used anti-hypertensive drugs. Between patients treated with RASBs and those with other anti-hypertensive drugs, patient survival rates were higher in patients treated with RASBs.

## 1. Introduction

Chronic kidney disease (CKD) is an important public health problem worldwide. It can progress to kidney failure, thus requiring renal replacement therapies, including hemodialysis (HD), peritoneal dialysis, and kidney transplantation. HD is the most commonly used modality among the three renal replacement therapies. As of 2021, the number of HD patients in South Korea was approximately 99,198. Hence, the proportion of HD among the three renal replacement therapies was approximately 78.1% [1]. Greater mortality is seen in HD patients than in the general population without CKD, and several factors, such as various comorbidities, uremic toxin, or dialysis, are associated with this high mortality; therefore, researchers have tried to identify the risk factors for mortality and the treatments to improve patient survival [2]. Despite these efforts, HD patients have greater mortality than those with other comorbidities, and there is insufficient evidence regarding the interventions used to improve prognosis in HD patients.

Renin-angiotensin system blockers (RASBs), including angiotensin-converting enzyme inhibitors and angiotensin II receptor blockers, are among the most commonly used anti-hypertensive drugs worldwide. Many studies have demonstrated the beneficial effects of RASB since captopril, the first RASB, was developed in 1975 [3]. Recent guidelines recommend the use of RASB as the first choice in patients with proteinuria and hypertension, regardless of the status of diabetes mellitus (DM) [4]. RASBs induce survival benefits in patients with myocardial infarction or heart failure [5,6]. However, there have been inconsistent results on patient survival benefits after the use of RASBs in patients on HD, despite proven survival benefits in patients with cardiovascular diseases and/or non-dialysis CKD [7,8,9,10,11,12]. Therefore, recent guidelines recommend the use of RASBs as the first choice among anti-hypertensive drugs in patients with non-dialysis CKD [4]. Additional studies are needed to confirm whether the use of RASBs induces survival benefits in patients on HD. This study aimed to evaluate patient survival after the use of RASBs in a large sample of maintenance HD patients.

## 2. Materials and Methods

### 2.1. Data Source and Study Population

This retrospective study used laboratory and clinical data from the national HD quality assessment program and claim data from the Health Insurance Review and Assessment Service (HIRA) of South Korea [13,14]. Briefly, the fourth, fifth, and sixth HD quality assessment programs were performed between July and December 2013, July and December 2015, and March and August 2018, respectively. The programs included maintenance HD patients (≥3 months), patients undergoing HD at least twice a week (≥8 per month), and patients aged ≥ 18 years. We analyzed the HD quality assessment data and the claim data of all HD patients who underwent HD quality assessment from the HIRA.

The numbers of patients included in the fourth, fifth, and sixth HD quality assessment programs were 21,846, 35,538, and 31,294, respectively. Among these patients, we excluded repeated participants or participants with insufficient datasets (n = 32,459) and those who underwent HD using a catheter (n = 1316). Finally, 54,903 patients were included in our study (Appendix A). The study was approved by the institutional review board of the Yeungnam University Medical Center (approval no. YUMC 2022-01-010). Informed consent was not obtained from the patients because of the retrospective study design and anonymization and de-identification of the records and information of the participants before the analysis.

### 2.2. Variables

Clinical data collected included age, sex, the underlying cause of end-stage renal disease, HD vintages (days), and the type of vascular access. Laboratory data collected during the assessment included hemoglobin (g/dL), Kt/V_urea_, serum albumin (g/dL), serum calcium (mg/dL), serum phosphorus (mg/dL), serum creatinine (mg/dL), pre-dialysis systolic blood pressure (mmHg), pre-dialysis diastolic blood pressure (mmHg), and ultrafiltration volume (L/session). These data were collected monthly, and all laboratory values were averaged from the monthly values. We calculated Kt/V_urea_ using the Daugirdas’ equation [15].

The codes for medications are shown in Appendix A. A patient using RASBs was defined as someone who had received more than one prescription during the 6 months of each HD quality assessment period. The patients were divided into three groups as follows: Group 1, no prescription for anti-hypertensive drugs; Group 2, prescription for anti-hypertensive drugs other than RASBs; and Group 3, prescription for RASBs. The use of aspirin, clopidogrel, ticlopidine, cilostazol, β-blockers, or statin as concomitant medication was also evaluated. If one or more prescriptions were identified for a year before the HD quality assessment program evaluation, the medication was considered used.

The presence of comorbidities was evaluated for a year before the HD quality assessment program. Comorbidity was defined using the codes utilized by Quan et al. [16,17]. The Charlson Comorbidity Index (CCI) included 17 comorbidities. All patients in our study underwent HD and were considered to have renal disease. Other comorbidities and their ICD-10 codes are shown in Appendix A. The CCI score was calculated after defining and identifying the comorbidities.

The patients were followed up until April 2022. If a patient was transferred for peritoneal dialysis or kidney transplantation, that date was considered as the end point of follow-up, and the data were censored. During the follow-up, clinical outcomes except death were defined using electronic data. The codes for censoring were O7072, O7071, or O7061 for peritoneal dialysis, and R3280 for kidney transplantation. Data regarding patient death were obtained from the HIRA database.

### 2.3. Statistical Analyses

Data were analyzed using the SAS Enterprise Guide version 7.1 (SAS Institute, Cary, NC, USA) or R version 3.5.1 (R Foundation for Statistical Computing, Vienna, Austria). Categorical variables were presented as numbers and percentages, whereas continuous variables were presented as means ± standard deviations. Pearson’s χ^2^ test or Fisher’s exact test was used to analyze the categorical variables. For continuous variables, means were compared using one-way analysis of variance, followed by the Tukey post hoc test. Survival estimates were calculated using the Kaplan–Meier curve and Cox regression analyses. The *p*-values for the comparison of survival curves were determined using the logrank test. Multivariate Cox regression analyses were adjusted for age, sex, the type of vascular access, the underlying cause of end-stage renal disease, CCI score, HD vintage, ultrafiltration volume, Kt/V_urea_, hemoglobin, serum albumin, serum creatinine, serum phosphorus, serum calcium, systolic blood pressure, diastolic blood pressure, and the use of aspirin or statin. Multivariate Cox regression analyses were performed using the enter mode. The proportional hazard assumption for the Cox proportional hazard model was tested using the Schoenfeld residuals test for continuous variables and log–log plots for categorical variables. All covariates were satisfied for proportionality assumption. We included all variables associated with patient’s survival as covariates in our analysis. Variables were selected from the patients’ baseline characteristics, which are well-known factors that affect patient survival outcomes. However, aspirin was selected among other antiplatelet agents because the proportions of patients who were prescribed aspirin were greater than those who were prescribed other antiplatelet agents.

We also performed analyses using propensity score weights. We created propensity score weights for the three groups using generalized boosted models for the following variables: age, sex, the underlying cause of end-stage renal disease, CCI score, HD vintage, ultrafiltration volume, Kt/V_urea_, hemoglobin, serum albumin, serum creatinine, serum phosphorus, serum calcium, systolic blood pressure, diastolic blood pressure, and the use of aspirin or statins. Statistical significance was set at *p* < 0.05.

## 3. Results

### 3.1. Clinical Characteristics

The number of patients in Group 1, Group 2, and Group 3 were 28,521, 9571, and 16,811, respectively (Table 1).

Group 2 patients were older in age and had experienced a greater duration of HD than those in the other two groups. Additionally, the dialysis adequacy was greater in Group 2 than in the other two groups. Male predominance was greater in Group 3 than in the other two groups. The proportion of DM and the use of aspirin or statin were greater in Group 3 than in the other two groups. The proportions of patients receiving other antiplatelet agents were lower in Group 1 than in the other groups, and the trends were similar in those receiving aspirin. However, the proportions of patients receiving aspirin were greater than those receiving other antiplatelet agents. Additionally, Group 3 had the highest CCI scores, ultrafiltration volumes, systolic blood pressures, and serum creatinine levels.

The numbers of patients with myocardial infarction or heart failure in younger and older patients were 14,183 (41.1%) and 10,749 (52.8%), respectively (*p* < 0.001). The number of patients with myocardial infarction or heart failure among DM or non-DM patients were 12,733 (52.7%) and 12,199 (39.7%), respectively (*p* < 0.001). The number of male and female patients with myocardial infarction or heart failure were 15,181 (46.3%) and 9751 (44.1%), respectively (*p* < 0.001). The proportions of patients with myocardial infarction and heart failure were greater among older, male patients and those with DM than among younger, female patients and non-DM patients.

### 3.2. Survival Analyses

At the follow-up end point, the number of patients in the survivor, death, peritoneal dialysis, and kidney transplantation subgroups were 17,073 (59.9%), 9433 (33.1%), 77 (0.3%), and 1938 (6.8%) in Group 1; 4273 (44.6%), 4501 (47.0%), 41 (0.4%), and 756 (7.9%) in Group 2; and 7778 (46.3%), 7499 (44.6%), 74 (0.4%), and 1460 (8.7%) in Group 3, respectively (*p* < 0.001).

The five-year survival rates in Groups 1, 2, and 3 were 72.1%, 64.5%, and 66.6%, respectively (Figure 1, *p* < 0.001 for Group 1 vs. Group 2 or 3; *p* = 0.001 for Group 2 vs. Group 3).

Group 1 had the highest patient survival rates among the three groups, and Group 3 had higher patient survival rates compared to Group 2. The univariate Cox regression analyses showed that the hazard ratios were 1.33 (95% CI, 1.29–1.38) in Group 2 and 1.25 (95% CI, 1.22–1.29) in Group 3 compared to that of Group 1 (Table 2).

The hazard ratio was 1.06 (95% CI, 1.03–1.10) for Group 2 compared to that of Group 3. Group 3 had higher patient survival rates than Group 2; however, the difference in patient survival rates between Group 2 and Group 3 was relatively small. Multivariate Cox regression analyses showed similar trends as those of univariate analyses.

We performed subgroup analyses according to the presence of myocardial infarction or heart failure, DM, age, and sex (Figure 2).

Analyses among patients without myocardial infarction or heart failure had trends similar to those of the total cohort. In patients with myocardial infarction or heart failure, the multivariate analyses were statistically significant between Group 1 and Group 2. Results from the older (≥65 years), DM, and female subgroups were similar to those of the total cohort. However, the multivariate Cox regression analyses were not statistically significant among the young and non-DM subgroups. Male patients in Group 1 had higher patient survival rates compared to the male patients in Group 2 and 3.

The numbers of patients who were prescribed β-blockers in Groups 2 and 3 were 5049 (52.8%) and 9241 (55.0%), respectively (*p* < 0.001). We reclassified patients in Groups 2 and 3 into the following four groups according to the use of RAS blockers and β-blockers: NR-NB, patients who were not on either of the two medications; NR-B, patients who were not on RAS blockers but were on β-blockers; R-NB, patients who were on RAS blockers but not on β-blockers; R-B, patients receiving both medications. The Kaplan–Meier curve showed that the 5-year survival rates in the NR-NB, NR-B, R-NB, and R-B groups were 64.2%, 64.8%, 66.4%, and 66.7%, respectively (Appendix A). Cox regression analyses are shown in Appendix A. The use of β-blockers did not improve patient survival in patients with or without the concomitant use of RAS blockers.

### 3.3. Analyses Using Propensity Score Weights

We compared the baseline characteristics and Kaplan–Meier survival rates using appropriate sampling weights. The balance among the three groups was evaluated using the maximum pairwise absolute standardized mean differences (ASMDs) of the covariates before and after weighting (Appendix A). After weighting, the maximum ASMDs and differences in baseline characteristics decreased for most covariates (Appendix A). The Kaplan–Meier curves generated using weighted data showed that the 5-year survival rates of Groups 1, 2, and 3 were 69.6%, 67.1%, and 68.6%, respectively (Appendix A). Group 1 had the best patient survival rate and Groups 3 had a better patient survival than did Group 2 (*p* < 0.001 for trend; *p* < 0.001 for Group 1 vs. Group 2 or 3; *p* = 0.001 for Group 2 vs. Group 3). However, the differences were lower in analyses using weighted data than in analyses using unweighted data. Cox regression analyses using weighted data are shown in Appendix A. Furthermore, the results from the analyses of weighted data were similar to those obtained using data from the entire cohort.

## 4. Discussion

We analyzed 54,903 patients who were evaluated in the HD quality assessment program in South Korea. Our results showed that Group 1 had the highest patient survival rates among the three groups. The Kaplan–Meier curves showed significantly higher patient survival rates in Group 3 compared to Group 2. The multivariate Cox regression analyses showed significant survival benefit in Group 3 compared to Group 2. These trends were also similar to those obtained from the analyses of patients who were older, female, had DM, and did not have myocardial infarction or heart failure. Additionally, we performed analyses using propensity score weights and the results were similar to those obtained using unweighted data.

In our study, the use of RASBs was associated with favorable patient survival rates compared to that of other antihypertensive drugs. Previous studies on the association between RASB use and patient survival among those on HD have shown inconsistent results [7,8,9,10,11,12]. Two randomized controlled trials did not show an association between RASB use and patient survival among patients on HD [7,8]. Zannad et al. enrolled 397 patients on HD with 24 months of follow-up. However, they showed that patients treated with Fosinopril did not have better outcomes, including each or composite outcomes compared to those not given Fosinopril [7]. Ruggenenti et al. enrolled 269 HD patients with hypertension and/or left ventricular hypertrophy, with 42 months of follow-up [8]. Their study also did not show favorable outcomes (cardiovascular death, non-fatal myocardial infarction, and non-fatal stroke) among patients treated with Ramipril. These two studies had a randomized controlled design; however, they were limited by low event rates and relatively short-term follow-up duration. Retrospective studies showed favorable results in patients treated with RASBs compared to randomized trials [9,10,11,12]. Efrati et al. analyzed 126 patients on HD and showed a 52% reduction in the risk of mortality in patients treated with RASBs compared to those not treated with RASBs [9]. McCullough et al. evaluated 368 dialysis patients admitted to a coronary care unit; they showed that the use of RASBs was associated with long-term patient survival [10]. Similarly, a previous study in patients with both acute myocardial infarction and end-stage renal disease also showed that the use of RASBs (n = 1025) was associated with a low mortality of 30 days (48% reduction in mortality compared to patients without RASB) [11]. Berger et al. analyzed data from the Dialysis Outcomes and Practice Patterns Study and enrolled 11,421 incident and 7124 prevalent HD patients [12]. They showed favorable outcomes in patients treated with RASBs in both HD groups. These retrospective studies were limited by their design, but had a longer-term follow-up durations and larger sample sizes than randomized studies. Our results were consistent with those of other retrospective studies. However, the survival benefits induced in patients treated with RASBs were relatively small. In the multivariate analyses, the increase in hazard ratio was 0.07 between Groups 2 and 3.

Our results showed that Group 1 had the highest patient survival rates among the three groups, and subgroup analyses among older patients and those with DM had similar trends to those of the total cohort. These results suggest that the survival benefits induced by RASBs may be limited to patients with clinical or subclinical cardiovascular diseases. Group 1 patients were not treated with anti-hypertensive drugs, and most of them may not have had hypertension as a comorbidity. The prevalence of clinical/subclinical cardiovascular disease might be lower in Group 1 than the other two groups treated with anti-hypertensive drugs. These differences may be associated with the highest patient survival rates in Group 1 among the three groups. Additionally, older or DM patients were associated with high cardiovascular disease compared to young or non-DM patients. In these populations with a high risk of cardiovascular disease, patients treated with RASBs were associated with better outcomes than those treated with other antihypertensive drugs. A previous study among DM patients who were not undergoing dialysis revealed that the use of RASBs improved left ventricular hypertrophy [18].

Survival trends among the three groups were different between the patients with and without myocardial infarction or heart failure. Results in patients without these heart diseases were similar to those of the total cohort. However, in patients with these heart diseases, a difference in survival rate was only observed between Groups 1 and 2. RAS blockers have been shown to be beneficial in the treatment of cardiovascular diseases among patients with various conditions. However, in our study, the survival benefits induced by RAS blockers were more prominent in patients without myocardial infarction or heart failure than in those with myocardial infarction or heart failure. Several hypotheses may be used to explain the non-association between the use of RAS blockers and survival benefits in patients on HD with myocardial infarction or heart failure. First, patients with heart disease have greater RAS activity than those without heart disease. Furthermore, patients on HD have greater RAS activity than those without chronic kidney disease [19,20]. The high RAS activity caused by these two factors is associated with the decreased efficacy of RAS blockers. Consequently, insufficient inhibition of RAS activity is associated with attenuated survival benefits of RAS blockers in patients on HD with heart disease. Second, patients with heart diseases are prone to developing hyperkalemia. Furthermore, the use of RAS blockers in such patients can be associated with a higher risk of hyperkalemia than those who do not use RAS blockers and/or have heart diseases [21]. A high risk of hyperkalemia is also associated with attenuated benefits of RAS blockers. Third, this may be associated with a high prevalence of intra-dialytic hypotension in patients with heart disease. Intradialytic hypotension in Group 2 and 3 patients treated with anti-hypertensive drugs might be more frequent than in Group 1. Nevertheless, the benefit of RASB may be associated with no significant difference in patient survival between Groups 1 and 3.

Our study included a large sample size. The HD quality assessment program was intended to improve the outcomes of maintenance HD patients. As a Korean government agency, the HIRA performs HD quality assessments in almost all HD facilities at regular intervals. Our study included approximately 46.0% of maintenance HD patients in South Korea at the time of the relevant HD quality assessment. Furthermore, our study included varied clinical and laboratory data, despite a representative sample and a relatively long-term follow-up compared to previous randomized trials.

In our study, the use of β-blockers did not improve patient survival in patients, irrespective of whether RAS blockers were used. It is well known that the use of β-blockers is associated with survival benefits in various populations [22,23,24,25,26]. However, evidence regarding these survival benefits are limited and conflicting in patients on HD [27]. Several factors can be associated with decreased survival benefits induced by β-blockers in patients on HD. First, different pharmacokinetics and cardio-selectivity can be associated with different survival outcomes and benefits. Atenolol and metoprolol are easily removed by HD; moreover, Weir et al. showed poorer clinical outcomes in patients receiving easily dialyzed medications than in those receiving poorly dialyzed medications [28]. Shireman et al. showed that cardio-selective β-blockers have better clinical outcomes than non-cardio-selective β-blockers [29]. In our study, β-blocker types were not evaluated and the patients on β-blockers may have included a mix of patients receiving easily and poorly dialyzed β-blockers, which may have induced difficulties in identifying the definite association between the use of β-blockers and patient survival. Second, patients on β-blockers may be prone to more advanced cardiovascular diseases, which may be associated with bias due to being less responsive to β-blockers. Third, a higher rate of adverse events, such as hypotension or bradycardia, is present in patients on HD than in those without HD [30]. Consequently, these factors may be associated with the absence of the survival benefits of receiving β-blockers among patients on HD in our study.

A previous study showed that male patients on HD had a higher prevalence of cardiovascular disease than did female patients on HD [31]. However, in our study, the survival benefits induced by RAS blockers were greater in female patients than in male patients. This may have been caused by the association between the use of RAS blockers and testosterone levels or erythropoietin resistance in male patients. Previous studies have shown that the use of RAS blockers is associated with a decrease in testosterone levels in male patients on HD, which could be associated with high erythropoietin resistance [32,33]. In addition, it is well known that low testosterone levels are per se a risk factor of mortality in male patients on HD [34,35]. Therefore, the survival benefits of RAS blockers in male patients may be attenuated by low testosterone levels combined with or without erythropoietin resistance.

Our study had some limitations. First, our study had a retrospective design. Second, comorbidities and the use of anti-hypertensive drugs were evaluated using claim data. A discrepancy between RASBs prescriptions and their real use may be present. Additionally, the etiology for the use of RASBs can influence patient outcomes. The effects of RASBs on patient survival may be different between patients who used RASB due to myocardial infarction or heart failure and those who used RASB for simple blood pressure control. Third, our study did not include data regarding the cause of death or heart function-related measurements such as the left ventricular hypertrophy, cardiac mass, or ejection fraction. The benefits of RASBs are mainly seen in cardiovascular diseases, and information on cardiovascular death would be useful in identifying the effect of RASBs beyond all-cause mortality. Additionally, heart function-related measurements would be essential to discriminate the presence/class of heart diseases because the effect of an RASB is clearer in patients with heart diseases. Additionally, these data would indicate whether the use of RASBs is beneficial in heart function beyond simple patient death. Fourth, many baseline characteristics among the three groups were different, and this may have led to selection bias and the confounding effects of these variables. However, we tried to attenuate these effects using multivariate analyses, subgroup analyses, and analyses using propensity score weights; the results of these analyses all showed similar trends.

In conclusion, our study showed the highest survival rates in patients who were not treated with anti-hypertensive drugs. Among patients treated with RASBs and those treated with other anti-hypertensive drugs, patient survival rates were higher in patients treated with RASBs. Our results should be carefully interpreted because of the study’s limitations. Further randomized prospective studies are needed to draw conclusions on whether the use of RASBs is associated with favorable outcomes in patients on HD.

## Figures and Tables

**Figure 1 jcm-12-03301-f001:**
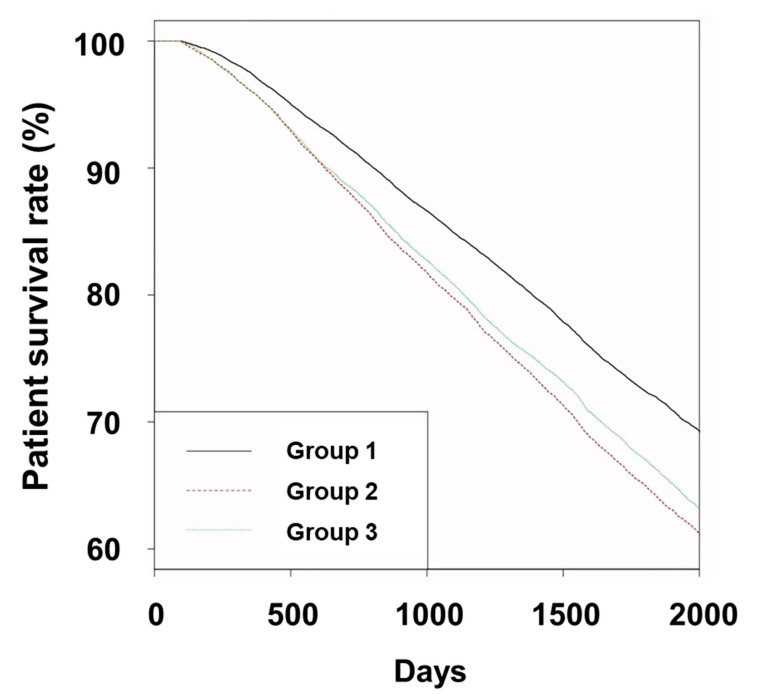
Kaplan–Meier curves for patient survival in groups. Abbreviations: Group 1, patients without prescription for any anti-hypertensive drugs; Group 2, patients with prescription for other anti-hypertensive drugs except renin-angiotensin system blockers; and Group 3, patients with prescription for renin-angiotensin system blockers.

**Figure 2 jcm-12-03301-f002:**
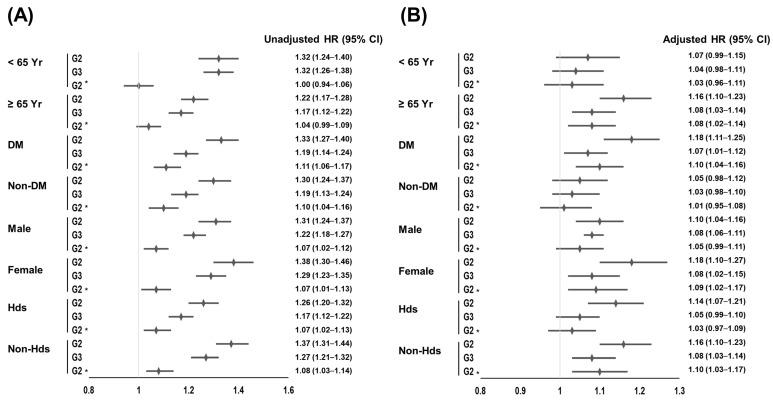
Forest plots for the HR and 95% CI according to subgroups. (**A**) Univariate Cox regression analysis; (**B**) multivariate Cox regression analyses. Adjustments were made according to age, sex, cause of end-stage renal disease, Charlson comorbidity index score, type of vascular access, hemodialysis vintage, ultrafiltration volume, Kt/V_urea_, hemoglobin, serum albumin, serum creatinine, serum phosphorus, serum calcium, systolic blood pressure, diastolic blood pressure, and the use of aspirin or statin. Reference group for HR was G1, except for G3, which was a reference group for G2*. Abbreviations: CI, confidence interval; DM, diabetes mellitus; G1, patients without prescription for any anti-hypertensive drugs; G2, patients with prescription for other anti-hypertensive drugs except renin-angiotensin system blockers; and G3, patients with prescription for renin-angiotensin system blockers; Hds, patients with myocardial infarction or heart failure; Non-Hds, patients without myocardial infarction or heart failure; HR, hazard ratio; Yr, years.

**Table 1 jcm-12-03301-t001:** Patient clinical characteristics.

	Group 1	Group 2	Group 3	*p*-Value
Age (years)	59.8 ± 13.4	61.3 ± 12.4 *	60.2 ± 12.5 *^+^	<0.001
Sex (male, %)	16,789 (58.9%)	5553 (58.0%)	10,442 (62.1%)	<0.001
Hemodialysis vintage (days)	1329 ± 1715	1954 ± 1782	1774 ± 1567 *^+^	<0.001
Underlying cause of ESRD				<0.001
Diabetes mellitus	11,940 (41.9%)	4089 (42.7%)	8122 (48.3%)	
Hypertension	7338 (25.7%)	2641 (27.6%)	4430 (26.4%)	
Glomerulonephritis	3098 (10.9%)	1049 (11.0%)	1657 (9.9%)	
Other	2717 (9.5%)	773 (8.1%)	1091 (6.5%)	
Unknown	3428 (12.0%)	1019 (10.6%)	1511 (9.0%)	
CCI score	7.4 ± 2.9	7.4 ± 2.9	7.7 ± 2.9 *^+^	<0.001
Follow-up duration (days)	1810 ± 810	1895 ± 929 *	1910 ± 917 *	<0.001
Type of vascular access				0.552
Arteriovenous fistula	24,353 (85.4%)	8126 (84.9%)	14,330 (85.2%)	
Arteriovenous graft	4168 (14.6%)	1445 (15.1%)	2481 (14.8%)	
Kt/V_urea_	1.53 ± 0.28	1.54 ± 0.27 *	1.52 ± 0.27 *	<0.001
Ultrafiltration volume (L/session)	2.20 ± 0.97	2.31 ± 0.93 *	2.38 ± 0.94 *^+^	<0.001
Hemoglobin (g/dL)	10.7 ± 0.8	10.7 ± 0.8 *	10.6 ± 0.8 *^+^	<0.001
Serum albumin (g/dL)	3.99 ± 0.33	3.98 ± 0.35	3.99 ± 0.34	0.072
Serum phosphorus (mg/dL)	4.95 ± 1.33	4.98 ± 1.41	5.00 ± 1.41 *	0.002
Serum calcium (mg/dL)	8.86 ± 0.80	8.98 ± 0.85 *	8.93 ± 0.84 *^+^	<0.001
Systolic blood pressure (mmHg)	138 ± 16	142 ± 15 *	144 ± 14.6 *^+^	<0.001
Diastolic blood pressure (mmHg)	78 ± 10	78 ± 9 *	78 ± 10 *	<0.001
Serum creatinine (mg/dL)	9.3 ± 2.8	9.5 ± 2.7 *	9.7 ± 2.7 *^+^	<0.001
Use of aspirin	10,327 (36.2%)	4735 (49.5%)	8544 (50.8%)	<0.001
Use of clopidogrel	3825 (13.4%)	1779 (18.6%)	3524 (21.0%)	<0.001
Use of ticlopidine	368 (1.3%)	241 (2.5%)	400 (2.4%)	<0.001
Use of cilostazol	1579 (5.5%)	790 (8.3%)	1272 (7.6%)	<0.001
Use of stain	7955 (27.9%)	2900 (30.3%)	5390 (32.1%)	<0.001

Data are expressed as means ± standard deviations for continuous variables and as numbers (percentages) for categorical variables. *p*-values were tested using one-way analysis of variance, followed by the Tukey post hoc test, and the Pearson’s χ^2^ test for categorical variables. Group 1, patients without prescription for any blood pressure-lowering drugs; Group 2, patients with prescription for other blood pressure-lowering drugs except renin-angiotensin system blockers; and Group 3, patients with prescription of renin-angiotensin system blockers. Abbreviations: CCI, Charlson comorbidity index; ESRD, end-stage renal disease. * *p* < 0.05 vs. No-prescription group, + *p* < 0.05 vs. Non-RASB group.

**Table 2 jcm-12-03301-t002:** Cox regression analyses for patient survival.

	Univariate		Multivariate	
	HR (95% CI)	*p*	HR (95% CI)	*p*
Group				
Ref: Group 1				
Group 2	1.33 (1.29–1.38)	<0.001	1.13 (1.08–1.18)	<0.001
Group 3	1.25 (1.22–1.29)	<0.001	1.07 (1.03–1.11)	<0.001
Ref: Group 3				
Group 2	1.06 (1.03–1.10)	0.001	1.06 (1.02–1.11)	0.009
Underlying cause of ESRD (ref: DM)	0.81 (0.80–0.82)	<0.001	0.90 (0.89–0.91)	<0.001
Age (increase per 1 year)	1.06 (1.06–1.06)	<0.001	1.06 (1.06–1.06)	<0.001
Type of vascular access (ref: arteriovenous fistula)	1.51 (1.46–1.56)	<0.001	1.19 (1.14–1.24)	<0.001
CCI score (increase per 1 score)	1.14 (1.13–1.14)	<0.001	1.07 (1.06–1.08)	<0.001
Sex (ref: male)	0.87 (0.84–0.89)	<0.001	0.74 (0.71–0.77)	<0.001
Hemodialysis vintage (increase per 1 day)	1.00 (1.00–1.00)	0.183	1.00 (1.00–1.01)	<0.001
Ultrafiltration volume (increase per 1 kg/session)	0.92 (0.90–0.93)	<0.001	1.07 (1.05–1.09)	<0.001
KtV_urea_ (increase per 1 unit)	0.91 (0.87–0.97)	0.001	0.81 (0.75–0.87)	<0.001
Hemoglobin (increase per 1 g/dL)	0.86 (0.85–0.88)	<0.001	0.90 (0.88–0.92)	<0.001
Serum albumin (increase per 1 g/dL)	0.37 (0.36–0.39)	<0.001	0.63 (0.59–0.66)	<0.001
Serum creatinine (increase per 1 mg/dL)	0.87 (0.86–0.87)	<0.001	0.93 (0.93–0.94)	<0.001
Serum phosphorus (increase per 1 mg/dL)	0.85 (0.84–0.86)	<0.001	1.04 (1.03–1.06)	<0.001
Serum calcium (increase per 1 mg/dL)	0.94 (0.92–0.95)	<0.001	1.06 (1.04–1.09)	<0.001
Systolic blood pressure (increase per 1 mmHg)	1.01 (1.01–1.01)	<0.001	1.01 (1.00–1.01)	<0.001
Diastolic blood pressure (increase per 1 mmHg)	0.98 (0.98–0.99)	<0.001	1.00 (1.00–1.01)	0.018
Use of aspirin	1.16 (1.13–1.19)	<0.001	0.96 (0.93–0.99)	0.010
Use of statin	1.10 (1.07–1.14)	<0.001	0.95 (0.92–0.99)	0.010

Multivariate analysis was adjusted according to age, sex, the underlying cause of ESRD, CCI score, type of vascular access, hemodialysis vintage, ultrafiltration volume, Kt/V_urea_, hemoglobin, serum albumin, serum creatinine, serum phosphorus, serum calcium, systolic blood pressure, diastolic blood pressure, and the use of aspirin or statin and was performed using enter mode. Group 1, patients without prescription for any blood pressure-lowering drugs; Group 2, patients with prescription for other blood pressure-lowering drugs except renin-angiotensin system blockers; and Group 3, patients with prescription for renin-angiotensin system blockers. Abbreviations: CCI, Charlson comorbidity index; CI, confidence interval; DM, diabetes mellitus; ESRD, end-stage renal disease; HR, hazard ratio.

## Data Availability

The raw data were generated at the Health Insurance Review and Assessment Service. The database can be requested from the Health Insurance Review and Assessment Service by sending a study proposal including the purpose of the study, study design, and duration of analysis via e-mail (turtle52@hira.or.kr) or through the website (https://www.hira.or.kr, assessed on 5 March 2023). The authors cannot distribute the data without permission.

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
