# Peer review of "Association of Renin-Angiotensin System Blockers with Survival in Patients on Maintenance Hemodialysis"

_jcm, 2023, doi:10.3390/jcm12093301_

Round 1

Reviewer 1 Report

Kang et al conducted a retrospective cohort study to evaluate patient survival with the use of RASBs in a large sample of maintenance HD patients. The authors found that compared to patients with prescription of anti-hypertensive drugs other than RASBs and prescription for RASBs, patients with no prescription for anti-hypertensive drugs showed highest survival rates. The manuscript is well written. However, I have two major comments for the authors to address:

1.     Comparator groups: The three groups are not comparable to each other. Patients not using any antihypertensive medications are healthier than patients using antihypertensive medications. Therefore, their survival is naturally expected to be better. To address unmeasured confounding, the authors should use IPTW or PS matching to make the groups as comparable as possible. They should also report the HRs before and after IPTW or PS matching.

2.     Multivariate Cox regression analyses: Did the authors test for proportionality assumption? If yes, please provide details in the results.

As  a minor comment, please provide sample selection flow diagram. 

Author Response

Kang et al conducted a retrospective cohort study to evaluate patient survival with the use of RASBs in a large sample of maintenance HD patients. The authors found that compared to patients with prescription of anti-hypertensive drugs other than RASBs and prescription for RASBs, patients with no prescription for anti-hypertensive drugs showed highest survival rates. The manuscript is well written. However, I have two major comments for the authors to address:

  1. Comparator groups: The three groups are not comparable to each other. Patients not using any antihypertensive medications are healthier than patients using antihypertensive medications. Therefore, their survival is naturally expected to be better. To address unmeasured confounding, the authors should use IPTW or PS matching to make the groups as comparable as possible. They should also report the HRs before and after IPTW or PS matching.

Answer: Thank you for your helpful comments and suggestions. We performed analyses using propensity score weights in line with your suggestion. We created propensity score weights for the three groups using generalized boosted models for the following variables: age, sex, underlying cause of end-stage renal disease, CCI score, HD vintage, ultrafiltration volume, Kt/Vurea, hemoglobin, serum albumin, serum creatinine, serum phosphorus, serum calcium, SBP, DBP, and use of aspirin or statins. We compared the baseline characteristics and Kaplan–Meier survival rates using appropriate sampling weights. The balance between the three groups was evaluated using the maximum pairwise absolute standardized mean differences (ASMDs) of the covariates before and after weighting (Figure S3). After weighting, the maximum ASMDs and differences in baseline characteristics decreased for most covariates (Table S4). The Kaplan–Meier curves generated using weighted data showed that the 5-year survival rates in Groups 1, 2, and 3 were 69.6%, 67.1%, and 68.6%, respectively (Figure S4). Group 1 had the best patient survival rate and Groups 3 had better patient survival than did Group 2 (P < 0.001 for trend; P < 0.001 for Group 1 vs. Group 2 or 3; P = 0.001 for Group 2 vs. Group 3). However, the differences were lower in analyses using weighted data than in analyses using unweighted data. Cox regression analyses using weighted data are shown in Table S5. Furthermore, the results from analyses of weighted data were similar to those obtained using data from the entire cohort.

We have added these details and supplementary data to the “Materials and Methods” and “Results” sections.

Figure S3. Absolute standardized difference plots for estimating propensity scores to generate weights. Left panel: effect size. Right panel: Kolmogorov–Smirnov statistics.

Table S4. Patients’ clinical characteristics after weighting.

Group 1

Group 2

Group 3

P

Age (years)

60.2 ± 0.1

60.5 ± 0.2

60.1 ± 0.1

0.124

Sex (male, %)

58.9%

58.7%

60.8%

0.004

Hemodialysis vintage (days)

1576 ± 12

1594 ± 19

1552 ± 16

0.396

Underlying cause of ESRD

0.029

Diabetes mellitus

43.9%

43.1%

45.1%

Hypertension

25.9%

26.4%

25.8%

Glomerulonephritis

10.4%

11.5%

10.6%

Others

8.5%

8.3%

7.9%

Unknown

11.2%

10.6%

10.5%

CCI score

7.5 ± 0.0

7.6 ± 0.0

7.6 ± 0.0

<0.001

Follow-up duration (days)

1849 ± 6

1855 ± 10

1863 ± 8

0.592

Type of vascular access

0.309

Arteriovenous fistula

84.7%

84.4%

85.2%

Arteriovenous graft

15.3%

15.6%

14.8%

Kt/Vurea

1.53 ± 0.00

1.53 ± 0.00

1.53 ± 0.00

0.864

Ultrafiltration volume (L/session)

2.26 ± 0.01

2.28 ± 0.01

2.30 ± 0.01

0.055

Hemoglobin (g/dL)

10.7 ± 0.0

10.7 ± 0.0

10.6 ± 0.0

0.171

Serum albumin (g/dL)

3.99 ± 0.00

3.99 ± 0.00

0.99 ± 0.00

0.944

Serum phosphorus (mg/dL)

4.97 ± 0.01

4.98 ± 0.02

4.98 ± 0.01

0.644

Serum calcium (mg/dL)

8.9 ± 0.0

8.9 ± 0.0

8.9 ± 0.0

0.096

Systolic blood pressure (mmHg)

141 ± 0

141 ± 0

142 ± 0

0.011

Diastolic blood pressure (mmHg)

78 ± 0

78 ± 0

78 ± 0

0.809

Serum creatinine (mg/dL)

9.4 ± 0.1

9.5 ± 0.1

9.5 ± 0.1

0.235

Use of aspirin

42.3%

43.6%

44.1%

0.032

Use of stain

29.1%

30.4%

30.4%

0.078

Data are expressed as mean ± standard error for continuous variables and as percentages for categorical variables.

Abbreviations: CCI, Charlson comorbidity index; ESRD, end-stage renal disease.

Figure S4. Kaplan–Meier curves of patient survival using weighted data according to the groups. The 5-year survival rates in Groups 1, 2, and 3 were 69.6%, 67.1%, and 68.6%, respectively (P < 0.001).

Table S5. Cox regression analyses for patient survival using weighted data

Univariate

Multivariate

HR (95% CI)

P

HR (95% CI)

P

Group

  Ref: Group 1

    Group 2

1.08 (1.06–1.11)

<0.001

1.13 (1.08–1.18)

<0.001

Group 3

1.04 (1.02–1.06)

<0.001

1.07 (1.03–1.11)

<0.001

  Ref: Group 3

Group 2

1.04 (1.02–1.06)

<0.001

1.06 (1.01–1.10)

0.013

Multivariate analysis was adjusted for age, sex, underlying cause of ESRD, CCI score, type of vascular access, hemodialysis vintage, ultrafiltration volume, Kt/Vurea, hemoglobin, serum albumin, serum creatinine, serum phosphorus, serum calcium, systolic blood pressure, diastolic blood pressure, and use of aspirin and statin, and was performed using the enter mode.

Abbreviations: CCI, Charlson comorbidity index; CI, confidence interval; ESRD, end stage renal disease; HR, hazard ratio.

  1. Multivariate Cox regression analyses: Did the authors test for proportionality assumption? If yes, please provide details in the results.

Answer: Thank you for your comments. In our study, the proportional hazard assumption for the Cox proportional hazard model was tested using the Schoenfeld residuals test for continuous variables or the log-log plots for categorical variables. All covariates were satisfied for proportionality assumption. We have added these details in the Methods section.

As a minor comment, please provide sample selection flow diagram. 

Answer: Thank you for your comments. As the reviewer suggested, we have added a new Figure showing the study flow diagram.

Reviewer 2 Report

This study assessed the survival benefit of RAS blockers in maintenance HD patients using a nationwide claims data. The methods and results are scientifically addressed, although I have a few comments.

1.     Methods:  The authors evaluated the use of aspirin as a concomitant medication. How about other anti-platelet medications, such as clopidogrel, ticlopidine, or cilostazol? I think a substantial portion of patients were on anti-platelet drugs other than aspirin.

2.     Methods: How were the variables chosen to be included in the multivariate Cox regression analysis?  Were they included because the authors chose them as a significant variable or because they showed a statistical significance in univariate analysis?

3.     Methods: There are many articles showing the survival benefits of beta-blockers in ESRD patients. In group 2, there must be patients under beta-blockers. I recommend the authors to adjust the effect of use of beta-blockers when comparing the survival between Group 2 and Group 3 in Kaplan-Meier survival analysis and in Cox regression analysis.

4.     Results: Table 1 – there is a typographical error in “nderlying cause of ESRD”.

5.     Discussion: on page 8, authors stated that “Additionally, older or DM patients were associated with high cardiovascular disease compared to young or non-DM patients”. Is this sentence addressed according to the results of this study? If not, I recommend to revise the sentence with a reference, because it can be understood that the results showed such findings.

In the subgroup analyses, females, older ages, DM patients, and those without heart disease showed a similar trend with the entire group.

6.     Discussion: authors should discuss about the subgroup analyses more specifically. Males are known to have higher cardiovascular disease and males were more prevalent in Group 3. Why did the survival benefits of RAS blockers were only seen in females?

7.     Discussion: authors continuously described that RAS blockers have cardiovascular benefits. However, the results showed a benefit in patients without heart disease but did not show a benefit in patients with heart disease. The authors shortly discussed about this by mentioning intra-dialysis hypotension. More detailed discussion about this finding is needed (why RAS blockers showed survival benefits in patients without heart disease but not in patients with heart disease).

Author Response

  1. Methods:  The authors evaluated the use of aspirin as a concomitant medication. How about other anti-platelet medications, such as clopidogrel, ticlopidine, or cilostazol? I think a substantial portion of patients were on anti-platelet drugs other than aspirin.

Answer: Thank you for your comments. In our study, the numbers of patients receiving clopidogrel, ticlopidine, or cilostazol were 3,825 (13.4%), 368 (1.3%), and 1,579 (5.5%) in Group 1; 1,779 (18.6%), 241 (2.5%), and 790 (8.3%) in Group 2; and 3,524 (21.0%), 400 (2.4%), and 1,272 (7.6%) in Group 3, respectively. The proportions of patients on antiplatelet agents were less in Group 1 than in the other groups, and the trends were similar in those on aspirin. However, the proportions of patients on aspirin were greater than those on other antiplatelet agents. We have added these details in the Methods and Results section.

  1. Methods: How were the variables chosen to be included in the multivariate Cox regression analysis?  Were they included because the authors chose them as a significant variable or because they showed a statistical significance in univariate analysis?

Answer: Thank you for your comments. We selected all variables associated with patients’ survival as covariates. All baseline characteristics are well-known factors associated with patient’s survival and were selected as covariates. However, we selected aspirin among antiplatelet agents because the proportion of patients who were prescribed aspirin was greater than those prescribed other antiplatelet agents. We have added these comments in the Methods section.

  1. Methods: There are many articles showing the survival benefits of beta-blockers in ESRD patients. In group 2, there must be patients under beta-blockers. I recommend the authors to adjust the effect of use of beta-blockers when comparing the survival between Group 2 and Group 3 in Kaplan-Meier survival analysis and in Cox regression analysis.

Answer: Thank you for your comments. The numbers of patients who were on β-blocker in Groups 2 and 3 were 5,049 (52.8%) and 9,241 (55.0%), respectively (P < 0.001). We reclassified patients in Groups 2 and 3 into the following four groups according to the use of RAS blockers and β-blockers: NR-NB, patients not on either of the two medications; NR-B, patients not on RAS blockers but were on β-blockers; R-NB, patients on RAS blockers but not on β-blockers; R-B, patients receiving both medications. The KaplanMeier curve showed that the 5-year survival rates in NR-NB, NR-B, R-NB, and R-B groups were 64.2%, 64.8%, 66.4%, and 66.7%, respectively (Figure S2). Cox regression analyses are shown in Table S3.

Figure S2. Kaplan–Meier curves of patient survival according the use of RAS blockers and β-blockers in Groups 2 and 3 (P = 0.007 for trend; P = 0.406 for NR-NB vs. NR-B; P = 0.012 for NR-NB vs. R-NB; P = 0.025 for NR-NB vs. R-B; P = 0.059 for NR-B vs. R-NB; P = 0.172 for NR-B vs. R-B; P = 0.479 for R-NB vs. R-B).

Abbreviations: RAS, renin-angiotensin system; NR-NB, patients not on either of the two medications; NR-B, patients not on RAS blockers but were on β-blockers; R-NB, patients on RAS blockers but not on β-blockers; R-B, patients receiving both medications.

Table S3. Cox regression analyses for patient survival according the use of RAS blockers and β-blockers in Groups 2 and 3

Univariate

Multivariate

HR (95% CI)

P

HR (95% CI)

P

Ref: NR-NB

  NR-B

0.97 (0.91–1.03)

0.336

1.06 (0.99–1.14)

0.112

  R-NB

0.92 (0.87–0.97)

0.002

0.93 (0.87–0.99)

0.027

  R-B

0.93 (0.88–0.98)

0.009

1.01 (0.95–1.07)

0.798

Ref: NR-B

  R-NB

0.94 (0.90–0.99)

0.033

0.88 (0.83–0.93)

<0.001

  R-B

0.96 (0.91–1.01)

0.111

0.95 (0.90–1.01)

0.104

Ref: R-NB

  R-B

1.02 (0.97–1.06)

0.477

1.09 (1.03–1.14)

0.002

Multivariate analysis was adjusted for age, sex, underlying cause of ESRD, CCI score, type of vascular access, hemodialysis vintage, ultrafiltration volume, Kt/Vurea, hemoglobin, serum albumin, serum creatinine, serum phosphorus, serum calcium, systolic blood pressure, diastolic blood pressure, and use of aspirin and statin, and was performed using enter mode.

Abbreviations: CCI, Charlson comorbidity index; CI, confidence interval; ESRD, end stage renal disease; HR, hazard ratio; RAS, renin-angiotensin system; NR-NB, patients not on either of the two medications; NR-B, patients not on RAS blockers but were on β-blockers; R-NB, patients on RAS blockers but not on β-blockers; R-B, patients receiving both medications.

In our study, the use of β-blockers did not improve patient survival in patients irrespective of whether RAS blockers were used. It is well known that the use of β-blockers is associated with survival benefits in various populations [1-5]. However, evidence regarding these survival benefits are limited and conflicting in patients on HD [6]. Several factors can be associated with decreased survival benefits of β-blocker in patients on HD. First, different pharmacokinetics and cardio-selectivity can be associated with different survival outcomes and benefits. Atenolol and metoprolol are easily removed by HD; moreover, Weir et al. showed poorer clinical outcomes in patients receiving easily dialyzed medications than in those receiving poorly dialyzed medications [7]. Shireman et al. showed that cardio-selective β-blocker have better clinical outcomes than non-cardio-selective β-blocker [8]. In our study, β-blocker types were not evaluated and patients on β-blocker may be mixed with patients receiving either easily or poorly dialyzed β-blockers (or cardio-selective or non-cardio-selective β-blockers), which may have induced difficulties in identifying the definite association between the use of β-blocker and patient survival. Second, patients on β-blocker may be prone to more advanced cardiovascular diseases, which may be association with bias due to lower responsive to β-blockers. Third, a higher rate of adverse events, such as hypotension or bradycardia, is present in patients on HD than in those without HD [9]. Consequently, these factors may be associated with the absence of the survival benefits of receiving β-blockers among patients on HD in our study. We have added these comments in the Results and Discussion section.

References

[1] Hjalmarson A, Goldstein S, Fagerberg B, Wedel H, Waagstein F, Kjekshus J, Wikstrand J, El Allaf D, Vítovec J, Aldershvile J, Halinen M, Dietz R, Neuhaus KL, Jánosi A, Thorgeirsson G, Dunselman PH, Gullestad L, Kuch J, Herlitz J, Rickenbacher P, Ball S, Gottlieb S, Deedwania P (March 2000). "Effects of controlled-release metoprolol on total mortality, hospitalizations, and well-being in patients with heart failure: the Metoprolol CR/XL Randomized Intervention Trial in congestive heart failure (MERIT-HF). MERIT-HF Study Group". JAMA. 2000;283(10): 1295-302.

[2] Leizorovicz A, Lechat P, Cucherat M, Bugnard F (February 2002). "Bisoprolol for the treatment of chronic heart failure: a meta-analysis on individual data of two placebo-controlled studies – CIBIS and CIBIS II. Cardiac Insufficiency Bisoprolol Study". American Heart Journal. 2002;143(2):301-7.

[3] Andersson C, Shilane D, Go AS, Chang TI, Kazi D, Solomon MD, et al. β-blocker therapy and cardiac events among patients with newly diagnosed coronary heart disease. Journal of the American College of Cardiology. 2014;64(3):247–52.

[4] Abbott KC, Trespalacios FC, Agodoa LY, Taylor AJ, Bakris GL. beta-Blocker use in long-term dialysis patients: association with hospitalized heart failure and mortality. Arch Intern Med. 2004 Dec 13-27;164(22):2465-71. 

[5] Heliste M, Pettilä V, Berger D, Jakob SM, Wilkman E. Beta-blocker treatment in the critically ill: a systematic review and meta-analysis. Ann Med. 2022 Dec;54(1):1994-2010.

[6] Tella A, Vang W, Ikeri E, Taylor O, Zhang A, Mazanec M, Raju S, Ishani A. β-Blocker Use and Cardiovascular Outcomes in Hemodialysis: A Systematic Review. Kidney Med. 2022 Apr 1;4(5):100460.

[7] Weir MA, Dixon SN, Fleet JL, Roberts MA, Hackam DG, Oliver MJ, Suri RS, Quinn RR, Ozair S, Beyea MM, Kitchlu A, Garg AX. β-Blocker dialyzability and mortality in older patients receiving hemodialysis. J Am Soc Nephrol. 2015 Apr;26(4):987-96. 

[8] Shireman TI, Mahnken JD, Phadnis MA, Ellerbeck EF. Effectiveness comparison of cardio-selective to non-selective β-blockers and their association with mortality and morbidity in end-stage renal disease: a retrospective cohort study. BMC Cardiovasc Disord. 2016 Mar 25;16:60.

[9] Badve SV, Roberts MA, Hawley CM, Cass A, Garg AX, Krum H, Tonkin A, Perkovic V. Effects of beta-adrenergic antagonists in patients with chronic kidney disease: a systematic review and meta-analysis. J Am Coll Cardiol. 2011 Sep 6;58(11):1152-61.

  1. Results: Table 1 – there is a typographical error in “nderlying cause of ESRD”.

Answer: Thank you for your helpful comments. We have corrected the error.

  1. Discussion: on page 8, authors stated that “Additionally, older or DM patients were associated with high cardiovascular disease compared to young or non-DM patients”. Is this sentence addressed according to the results of this study? If not, I recommend to revise the sentence with a reference, because it can be understood that the results showed such findings.

Answer: Thank you for your comments and suggestion. In our study, the numbers of patients with myocardial infarction or heart failure in young or older patients were 14,183 (41.1%) and 10,749 (52.8%), respectively (P < 0.001). The number of patients with myocardial infarction or heart failure among DM or non-DM patients were 12,733 (52.7%) and 12,199 (39.7%), respectively (P < 0.001). The number of male and female patients with myocardial infarction or heart failure were 15,181 (46.3%) and 9,751 (44.1%), respectively (P < 0.001). The proportion of patients with myocardial infarction or heart failure were greater among older, male patients or those with DM than patients with young age, female patients or non-DM patients. We have added these comments in the Results section.

In the subgroup analyses, females, older ages, DM patients, and those without heart disease showed a similar trend with the entire group.

  1. Discussion: authors should discuss about the subgroup analyses more specifically. Males are known to have higher cardiovascular disease and males were more prevalent in Group 3. Why did the survival benefits of RAS blockers were only seen in females?

Answer: Thank you for your comments. Previous study showed that male patients on HD had a higher prevalence of cardiovascular disease than did female patients on HD [1]. However, in our study, the survival benefits of RAS blockers were greater in female patients than in male patients. This may have been caused by the association between the use of RAS blockers and testosterone levels or erythropoietin resistance in male patients. Previous studies have shown that the use of RAS blockers is associated with a decrease in testosterone levels in male patients on HD, which could be associated with high erythropoietin resistance [2,3]. In addition, it is well known that low testosterone level per se is a risk factor of mortality in male HD patients [4,5]. Therefore, the survival benefits of RAS blockers in male patients may be attenuated by low testosterone levels combined with or without erythropoietin resistance. We have added these details in the Discussion section.

References

[1] Carrero JJ, de Jager DJ, Verduijn M, Ravani P, De Meester J, Heaf JG, Finne P, Hoitsma AJ, Pascual J, Jarraya F, Reisaeter AV, Collart F, Dekker FW, Jager KJ. Cardiovascular and noncardiovascular mortality among men and women starting dialysis. Clin J Am Soc Nephrol. 2011 Jul;6(7):1722-30.

[2] Koshida H, Takeda R, Miyamori I. Lisinopril decreases plasma free testosterone in male hypertensive patients and increases sex hormone binding globulin in female hypertensive patients. Hypertens Res. 1998 Dec;21(4):279-82.

[3] DeLong M, Logan JL, Yong KC, Lien YH. Renin-angiotensin blockade reduces serum free testosterone in middle-aged men on haemodialysis and correlates with erythropoietin resistance. Nephrol Dial Transplant. 2005 Mar;20(3):585-90. 

[4] Carrero JJ, Qureshi AR, Parini P, Arver S, Lindholm B, Bárány P, Heimbürger O, Stenvinkel P. Low serum testosterone increases mortality risk among male dialysis patients. J Am Soc Nephrol. 2009 Mar;20(3):613-20.

[5] Gungor O, Kircelli F, Carrero JJ, Asci G, Toz H, Tatar E, Hur E, Sever MS, Arinsoy T, Ok E. Endogenous testosterone and mortality in male hemodialysis patients: is it the result of aging? Clin J Am Soc Nephrol. 2010 Nov;5(11):2018-23.

  1. Discussion: authors continuously described that RAS blockers have cardiovascular benefits. However, the results showed a benefit in patients without heart disease but did not show a benefit in patients with heart disease. The authors shortly discussed about this by mentioning intra-dialysis hypotension. More detailed discussion about this finding is needed (why RAS blockers showed survival benefits in patients without heart disease but not in patients with heart disease).

Answer: Thank you for your comments. We have added some details regarding the non-association between the use of RAS blockers and survival benefits in patients with heart disease. RAS blockers have been shown to be beneficial in the treatment of cardiovascular diseases among patients with various conditions. However, in our study, the survival benefits of RAS blockers were more prominent in patients without myocardial infarction or heart failure than in those with without myocardial infarction or heart failure. Several hypotheses may be used to explain the non-association between the use of RAS blockers and survival benefits in patients on HD with myocardial infarction or heart failure. First, patients with heart disease have greater RAS activity than those without heart disease. Furthermore, patients on HD have greater RAS activity than those without chronic kidney disease [1,2]. High RAS activity caused by these two factors is associated with decreased efficacy of RAS blockers. Consequently, insufficient inhibition of RAS activity is associated with attenuated survival benefits of RAS blockers in patients on HD with heart disease. Second, patients with heart diseases are prone to develop hyperkalemia. Furthermore, the use of RAS blockers in such patients can be associated with higher risk of hyperkalemia than those without RAS blockers and/or without heart diseases [3]. High risk of hyperkalemia is also associated with attenuated benefits of RAS blockers. We have added these details in the Discussion section.

Reference

[1] Kimura G, Takahashi N, Kawano Y, Inenaga T, Inoue T, Nakamura S, Inoue T, Matsuoka H, Omae T. Plasma renin activity in hemodialyzed patients during long-term follow-up. Am J Kidney Dis. 1995 Apr;25(4):589-92.

[2] Malik U, Raizada V. Some Aspects of the Renin-Angiotensin-System in Hemodialysis Patients. Kidney Blood Press Res. 2015;40(6):614-22.

[3] Fudim M, Grodin JL, Mentz RJ. Hyperkalemia in Heart Failure: Probably Not O"K". J Am Heart Assoc. 2018 May 22;7(11):e009429.

Round 2

Reviewer 1 Report

NO comments.

Reviewer 2 Report

Authors have revised the manuscript appropriately. I have no further comments.